# Nickase fidelity drives EvolvR-mediated diversification in mammalian cells

Juan E. Hurtado[1], Adam J. Schieferecke [2,3], Shakked O. Halperin [1], John Guan[1], Dylan Aidlen[2], David V. Schaffer [1,2,3,4,5] ✉ & John E. Dueber [1,3,6] ✉

In vivo genetic diversifiers have previously enabled efficient searches of genetic variant fitness landscapes for continuous directed evolution. However, existing genomic diversification modalities for mammalian genomic loci exclusively rely on deaminases to generate transition mutations within target loci, forfeiting access to most missense mutations. Here, we engineer CRISPR-guided error-prone DNA polymerases (EvolvR) to diversify all four nucleotides within genomic loci in mammalian cells. We demonstrate that EvolvR generates both transition and transversion mutations throughout a mutation window of at least 40 bp and implement EvolvR to evolve previously unreported drug-resistant *MAP2K1* variants via substitutions not achievable with deaminases. Moreover, we discover that the nickase's mismatch tolerance limits EvolvR's mutation window and substitution biases in a gRNA-specific fashion. To compensate for gRNA-to-gRNA variability in mutagenesis, we maximize the number of gRNA target sequences by incorporating a PAM-flexible nickase into EvolvR. Finally, we find a strong correlation between predicted free energy changes underlying R-loop formation and EvolvR's performance using a given gRNA. The EvolvR system diversifies all four nucleotides to enable the evolution of mammalian cells, while nuclease and gRNA-specific properties underlying nickase fidelity can be engineered to further enhance EvolvR's mutation rates.

Since the late 20th century, researchers have engineered broadly impactful biotechnologies by hypermutating genes of interest and screening the resulting populations of genetic variants to isolate those with enhanced biomolecular function, a method since referred to as directed evolution.

Directed evolution classically begins with the diversification of genes encoding biomolecules of interest in vitro, typically by using error-prone PCR[1], DNA shuffling[2], or saturation mutagenesis[3]. These genes are stably introduced into cells or viruses either as replicating extrachromosomal elements or genomically-integrated transgenes, such that a given variant's cDNA sequence is retrievable by harvesting

the cells' DNA. Upon expression of the genetic library, selective pressures are applied to the host population to either eliminate dysfunctional variants or enhance the persistence or reproduction of functional variants to the degree that they execute the desired function. Such functionalities have included enhanced or gain-of-function catalytic activity in enzymes for biomanufacturing[4]; specific and high-affinity binding of proteins to ligands[5,6], proteins[7], or nucleic acids[8] for research and therapeutic applications; and improved infectivity, tissue-specific tropism, or manufacturing of engineered viral vectors for the delivery of gene therapies to target cells[9–11]. Advantageously, iterative cycles of diversification and selection for desired functions

[1]Department of Bioengineering, University of California, Berkeley, Berkeley, CA, USA. [2]Department of Molecular and Cell Biology, University of California, Berkeley, Berkeley, CA, USA. [3]QB3, University of California, Berkeley, Berkeley, CA, USA. [4]Department of Chemical and Biomolecular Engineering, University of California, Berkeley, Berkeley, CA, USA. [5]Helen Wills Neuroscience Institute, University of California, Berkeley, Berkeley, CA, USA. [6]Biological Systems & Engineering Division, Lawrence Berkeley National Laboratory, Berkeley, CA, USA. ✉e-mail: schaffer@berkeley.edu; jdueber@berkeley.edu

can be executed without prior knowledge of the mechanisms underlying these functionalities. As such, directed evolution is a powerful technique that enables the engineering and fine-tuning of incompletely understood biological properties, such as cellular drug resistance or cell tropism of viral vectors.

Despite its historically successful implementation for engineering biotechnologies, conventional directed evolution's reliance on discrete steps of library diversification and delivery of the library into a host population limits the diversity of variants that can be tested in each round to the number of host cells that can be transformed, and multiple rounds of directed evolution are labor-intensive as well as limiting for both the number of genes of interest that can be evolved in parallel and the practical duration of directed evolution campaigns. To eliminate diversity bottlenecks related to the transformation of genetic hosts and to improve the scalability of directed evolution campaigns, researchers have developed technologies enabling in vivo continuous evolution of target genes by recruiting error-prone polymerases[12–16] or base editors[17–20] to genes of interest directly within host cells, as we have recently reviewed[21]. Continuous genetic diversifiers eliminate bottlenecks arising from the transformation of libraries, while continuous and simultaneous diversification and selection allow early "hits" to seamlessly accumulate additional gain-of-function mutations, allowing unperturbed exploration of arbitrarily distant fitness peaks for as long as the diversifier remains active.

Nevertheless, the directed evolution of desired biomolecular properties requires selective pressures that faithfully interrogate these properties. Though activities such as protein-protein binding and enzyme catalysis can depend relatively little on host cell biology, other biological processes of great biotechnological interest can only be interrogated under species and cell type-specific biological contexts. Examples include the signaling kinetics of transmembrane receptors, such as GPCRs and synthetic T cell receptors[22,23], as well as orthogonal receptor-ligand signaling[24]. Accordingly, to engineer phenotypes relying on human cell biology via directed evolution, researchers have developed continuous diversifiers that enable the targeted hypermutation of genes of interest directly within human cell lines. Existing targeted hypermutators for in vivo continuous evolution in mammalian cells consist of either orthogonal viral polymerases[15,14]and replicases[16] that diversify genes of interest in viral genomes, or CRISPR-guided nucleobase deaminases that chemically mutate C or A nucleotides near or within gRNA target sites. However, neither of these modalities is completely suitable for diversifying all four nucleotides at endogenous mammalian genomic loci. Though orthogonal viral error-prone polymerases (AdPol, VEGAS) and replicases (REPLACE) can in principle diversify all nucleotides within genes of interests, the necessary use of viral infection and replication as a means of diversification and selection limits the use of AdPol, VEGAS, and REPLACE to applications in which the biochemical functionality under evolution can be coupled to viral or replicase propagation. As a result, assessing functional variants under transcriptionally-normalized and clinically-relevant expression strengths or investigating functional variation arising from perturbation of splicing and regulatory elements is not feasible with virus-based continuous evolution systems. Moreover, directed evolution campaigns with VEGAS, specifically, have been reported to be confounded by cheater variants[25]. In contrast, CRISPR-targeted nucleobase deaminases (CRISPR-X, dCas9-AIDx, HACE)[18,19,26] can efficiently mutate mammalian genomic loci directly within windows of 50 (CRISPR-X, dCas9-AIDx) or thousands of base pairs (HACE), but their reliance on deaminases critically limits the substitutions that can be directly accessed to transition mutations (C to T, A to G, G to A, T to C), forfeiting reliable access to the vast majority of missense mutations (Supplementary Fig. 1). Therefore, a technology capable of diversifying all 4 nucleotides within mammalian genomic loci to generate all 12 substitutions would enable access to previously untapped diversity spaces during in vivo continuous directed evolution campaigns in mammalian cells.

CRISPR-guided DNA polymerases, which we have previously referred to as "EvolvR,"[13] localize error-prone DNA synthesis to a user-defined locus by generating a priming nick at a target site. Typically, a D10A mutant Cas9 nickase that cuts only the gRNA target strand (nCas9) is fused to an error-prone E. coli Pol I and is directed by a gRNA to generate a single-stranded break at a target locus. After nCas9 dissociation, Pol I is believed to utilize the nicked strand's exposed 3' end as a primer for low-fidelity DNA synthesis, displacing the incumbent strand as it generates substitution mutations (Fig. 1a). Substitutions are "locked in" upon cell division or upon evasion of mismatch repair, generating new library members. As with CRISPR-guided deaminases, EvolvR's capacity to diversify genes directly within the native genome eliminates screening bottlenecks arising from inefficiencies in library delivery to host cells. However, unlike CRISPR-guided deaminases, EvolvR leverages error-prone DNA synthesis for mutagenesis and has been shown to diversify all four nucleotides in E. coli and S. cerevisiae[13,27].

Here, we engineer EvolvR to generate diversity in mammalian cells with comparable efficiency and window lengths per gRNA as those reported in microbes, and we apply EvolvR to identify drug-resistant MAP2K1 variants in A375 melanoma cells generated via transversion mutations. We also hypothesize that nickase and gRNA properties may influence EvolvR's mutation outcomes and thereafter discover that EvolvR's mutation window and substitution biases are limited by mismatch tolerance, as determined both by the gRNA sequence and a given Cas9 variant's biochemical properties. To compensate for gRNA-dependent variability in EvolvR's performance, we show that an EvolvR using a high-fidelity, PAM-flexible Cas9 ortholog drastically increases the number of gRNAs that can be used to functionally target EvolvR to a given locus. Finally, we use a PAM-flexible EvolvR to elucidate a strong correlation between the free energy change of R loop formation for a given gRNA and EvolvR's on-target substitution efficiency.

This work presents the design of a single biomolecule for diversifying all four nucleotides efficiently within native genomic loci in mammalian cells and improves upon EvolvR's original design by enabling PAM-flexible (NNG) targeting. Accordingly, the availability of potential target loci is considerably increased, enabling the diversification of virtually any position within the human genome.

## Results
### EvolvR diversifies mammalian genomic loci
To test whether CRISPR-guided DNA polymerases (henceforth referred to as "EvolvR") can diversify targeted genomic loci in human cells, we designed a modified version of the EvolvR construct we originally described composed of a nickase derived from an engineered SpCas9 variant (K848A, K1003A, R1060A) containing an additional RuvC-domain-inactivating D10A missense mutation fused to one of two variants of E. coli polymerase I harboring three (D424A, I709N, A759R) or five (D424A, I709N, A759R, F742Y, P796H) missense mutations that increase the polymerase's error rate, named PolI3M and PolI5M, respectively[13]. To test EvolvR in mammalian cells, we flanked enCas9 with two nuclear localization sequences along with an mCherry fluorescent reporter tag (Supplementary Fig. 2a). Additionally, we tested a version of PolI5M without the N-terminal flap endonuclease domain (PolI5MΔ) to determine whether the remaining Klenow fragment alone is sufficient for EvolvR-mediated mutagenesis.

A fluorescent reporter for gene editing was used to rapidly and sensitively measure the frequency of single substitution variants generated by EvolvR. Specifically, we transfected HEK293 cells expressing a genomically-integrated, single-copy of a blue fluorescent protein (BFP) gene with a plasmid to drive strong expression of EvolvR (Supplementary Fig. 2b) along with a 20 nt-long gRNA (henceforth referred to as "gBFP15") directing a nick 15 bp upstream

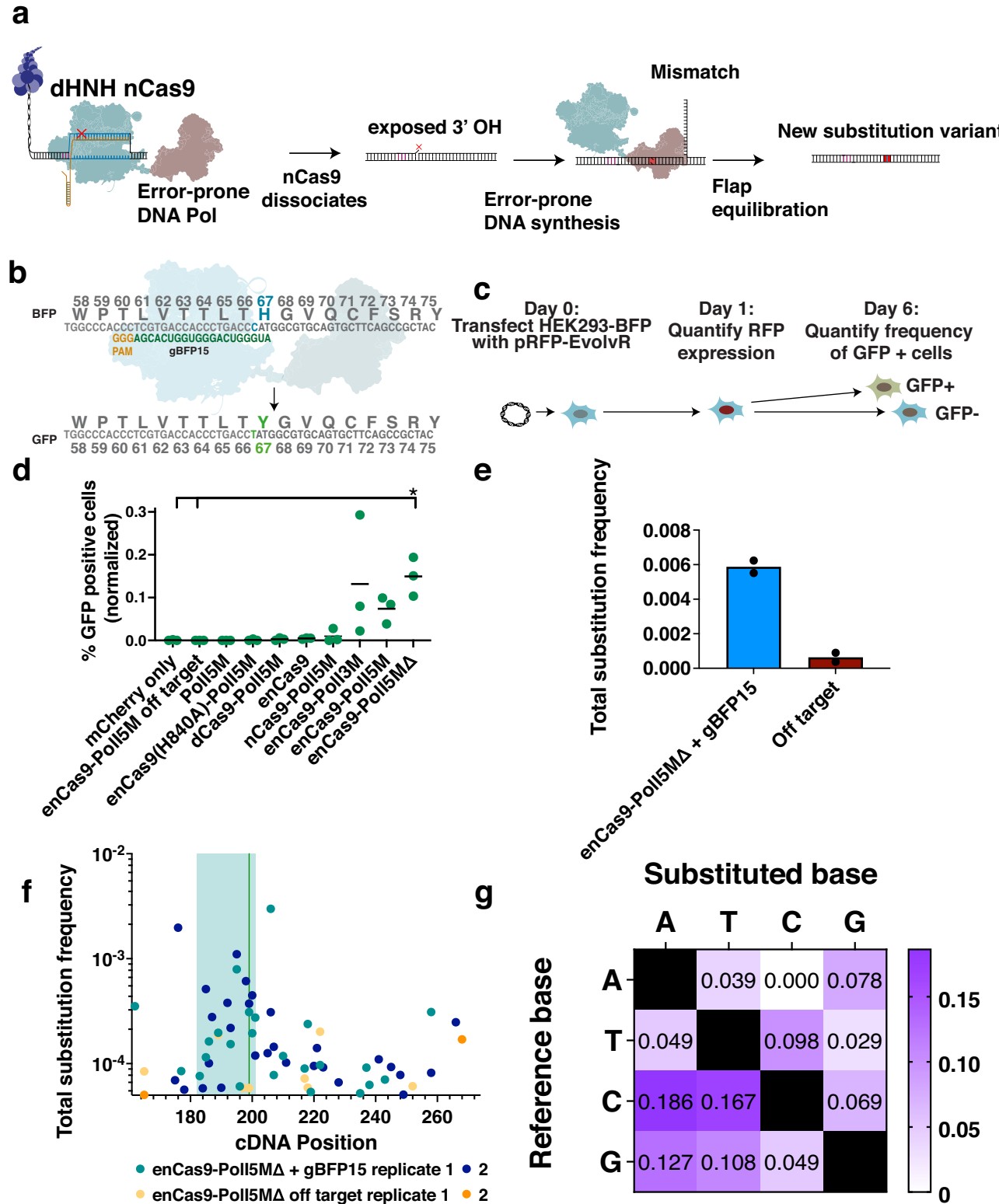

of a position that, upon undergoing an H67Y missense mutation resulting from replacing C199 with a T (henceforth referred to as 199 C > T), results in a transition from blue to green fluorescence (Fig. 1b). Six days after transfection, we measured the frequency of GFP positive cells by flow cytometry to compare the mutagenicity of different versions of EvolvR to controls (Figs. 1c, Supplementary Fig. 3). As expected, conditions in which enCas9 and PolI5M were not fused to one another, or targeted to the BFP locus, did not

significantly increase the frequency of GFP positive cells when compared to cells expressing mCherry alone (Fig. 1d). Additionally, EvolvR constructed with wildtype nCas9 fused to PolI5M did not consistently generate GFP positive cells. In contrast, all three EvolvR variants consisting of enCas9 fused to PolI3M, PolI5M, or PolI5MΔ consistently generated GFP positive cells in all replicates. Notably, EvolvR lacking the flap endonuclease domain generated the highest mean frequency of GFP-expressing cells in the final population,

**Fig. 1 | EvolvR enables targeted diversification of mammalian genomic loci.**
**a** EvolvR consists of a Cas9 nickase (nCas9) fused to an error-prone DNA polymerase. First, nCas9 generates an RNA-guided single-stranded break in the gRNA target strand. After nCas9 dissociation, the DNA polymerase performs error-prone strand-displacing DNA synthesis while generating substitutions errors. **b** Schematic depicting a fluorescent reporter gene for EvolvR mutagenesis whereby substitution 199 C > T results in missense mutation H67Y yielding expression of GFP. EvolvR cartoon is adapted from Halperin et al. 2018[13]. **c** BFP-expressing HEK293 cells are transfected with a plasmid encoding expression of mCherry-tagged EvolvR. Transfection efficiency is quantified to enable normalization of the final frequency of GFP positive cells by the frequency of cells expressing EvolvR on day 1. On day 6, cells are analyzed by flow cytometry to quantify the frequency of GFP-expressing cells. **d** Green dots represent the percentage of GFP positive cells normalized by transfection efficiency in three separate biological replicates generated by

constructs co-expressed with a gRNA targeting a nick 15 bp upstream of C199. "Off target" indicates the expression of a gRNA targeting safe harbor locus AAVS1. Asterisk indicates *p* values of less than 0.0328 and 0.0318 for comparisons with the mCherry only and off target controls, respectively, as determined by a two-way ANOVA and Tukey's HSD test. **e, f** EvolvR generates substitutions downstream of the nick as measured by amplicon sequencing of BFP. The total frequency of substitutions was quantified 40 bp from the nick (**f**) and at each position within BFP (**g**) in two biological replicates. Positional coordinates are in reference to the BFP transgene's forward cDNA sequence in the 5' to 3' direction. The blue rectangle marks the the gRNA target sequence window. The green vertical line marks the position of the 199 C > T substitution conferring expression of GFP. **g** The proportion of each substitution type observed downstream of the nick was normalized by the relative proportion of each reference nucleotide's appearance within the target strand.

though no differences in GFP positive cells generated between enCas9-PolI3M, enCas9-PolI5M, or enCas9-PolI5MΔ were statistically significant. These data suggest that Pol I's flap endonuclease domain is inessential for EvolvR's targeted mutagenicity in HEK293 cells, potentially due to redundancy with endogenous nuclear flap exonucleases.

Having found that EvolvR elevates the mutation rate of a specific nucleotide within a user-defined genomic locus, we quantified the diversity generated at the target locus (i.e. EvolvR's mutation window) using next generation sequencing. We again transfected HEK293 BFP cells with plasmids encoding mCherry-tagged enCas9-PolI5MΔ and gBFP15 or a gRNA targeting safe harbor control locus AAVS1 (an "off target" gRNA). To ensure normalization of substitution frequencies across samples and exclude untransfected cells from our analysis, cells expressing EvolvR were isolated by FACS by gating for mCherry fluorescence. One week after this sort, we harvested the genomic DNA of biological duplicates from each condition and quantified the frequency of substitutions at each position using next generation sequencing of BFP amplicons. To minimize the influence of base calling errors on variant analysis, we filtered all paired end reads for perfectly matching sequences within the overlapping region and filtered base calls by quality score 30 before subtracting the mean frequency of each possible variant in an untransfected negative control from the frequency of the corresponding variant in all experimental conditions. Variants with post-subtraction frequencies of less than 1/100,000 were then removed from subsequent analysis. Coexpression of EvolvR and gBFP15 significantly elevated the substitution rate throughout the gRNA spacer and beyond compared to the off target negative control (Fig. 1e, f). EvolvR's elevated substitution rates extended from the nick to the end of the region quantified, though the frequency of substitutions was markedly higher 40 bp downstream of the nick. All four nucleotides were diversified, and all but one of twelve possible substitutions (A to C) were represented in the set of background-subtracted variants (Fig. 1g), suggesting that EvolvR leverages error-prone DNA synthesis to access both transition and transversion substitutions.

Notably, though EvolvR generated mutations asymmetrically in the direction expected during error-prone DNA synthesis, mutations in the opposite direction upstream of the nick were also detectable. We and others previously reported this bidirectionality in mutagenesis in *S. cerevisiae*[27,28]. We postulate that bidirectional mutagenesis occurs due to Pol I participation in DNA repair processes initiated during replication fork collisions with EvolvR-generated single-stranded breaks. Specifically, we suggest that Pol I initiates error-prone transcription from free 3' ends remaining during MRE-mediated resection of double-stranded breaks, which are known to occur during replication fork collapse arising from single-stranded breaks[29,30]. This explanation is supported by recent work demonstrating that a Cas9-PolI fusion participated in generating templated insertions from double-stranded breaks[31].

## EvolvR-based screen identifies drug-resistant MAP2K1 exon 6 variants

Having demonstrated that EvolvR diversifies all four nucleotides within at least 40 bp of the nickase's cut site, we tested EvolvR's capacity for generating gain of function variants within native genomic loci by targeting EvolvR to *MAP2K1*(Fig. 2a), a gene for which variants have been previously identified to confer resistance to ERK pathway inhibitors in various cancers[32] and for which previous genomic diversifiers have successfully evolved drug-resistant variants[26,33]. To test EvolvR's capacity to evolve drug-resistant *MAP2K1*, we transiently transfected A375 cells with a plasmid encoding enCas9-PolI5M and one of three gRNAs targeting three different loci along *MAP2K1* exon 6, where mutations that confer resistance to selumetinib have been previously described[34]. After transfection, cells were split into three separate cultures per condition and allowed to recover for six days before addition of 1 μM selumetinib (Fig. 2b). Non-target control cell cultures transfected with EvolvR and a gRNA targeting a nonexistent BFP gene (gBFP45) did not noticeably replicate and in one of three replicates were not sufficiently viable after passaging into selective media for harvesting DNA. In contrast, cultures transfected with EvolvR and three gRNAs targeting three nonoverlapping loci in exon 6 contained visible colonies at the end of the selection period (Supplementary Fig. 4a).

Amplicon sequencing of *MAP2K1* gDNA from cells transfected with EvolvR revealed seven variants, six of which were substitution variants, that were enriched by at least 10-fold over the mean frequencies of those variants in a triplicate untreated, unselected control (Fig. 2c, Supplementary Figs. 4b, c). To validate the three most enriched drug-resistant *MAP2K1* variants (V211H, G213C Q214P, +S212), we cloned each and transiently transfected HEK293T cells with the resulting expression plasmids. MAP2K1-ERK activity was measured in the presence of selumetinib using a plasmid with an SRE-controlled luciferase reporter, previously used for measuring MAP2K1 activity in the presence of an inhibitor[33] (Fig. 2d). As positive and negative controls, we tested the discovered variants against known selumetinib-resistant variant V211D[35] and a plasmid encoding wildtype MAP2K1, respectively. V211H, G213C/Q214P, and +S212 all showed statistically significantly higher activity at all concentrations of selumetinib tested compared to wildtype MAP2K1. Additionally, we generated V211H in combination with G213C/Q214P and found that this variant's relative MAP2K1-ERK activity was ~2.3-fold higher than that of V211D and 204.7-fold higher than wildtype MAP2K1 under inhibition by 1 μM selumetinib. We attribute the absence of V211H, G213C, and Q214P triple missense variants in our selection to insufficient sampling of the quintuple substitution variant space with our culture size of ~$10^7$ cells. To our knowledge, all six enriched missense variants have not previously been described and were all generated via substitutions not accessible with deaminases, revealing EvolvR's potential to access diversity spaces yet unexplored by current in vivo diversifiers. Moreover, though A to C substitutions were absent from our initial quantification of EvolvR's substitution profile in BFP (Fig. 1g), the presence of an A to C

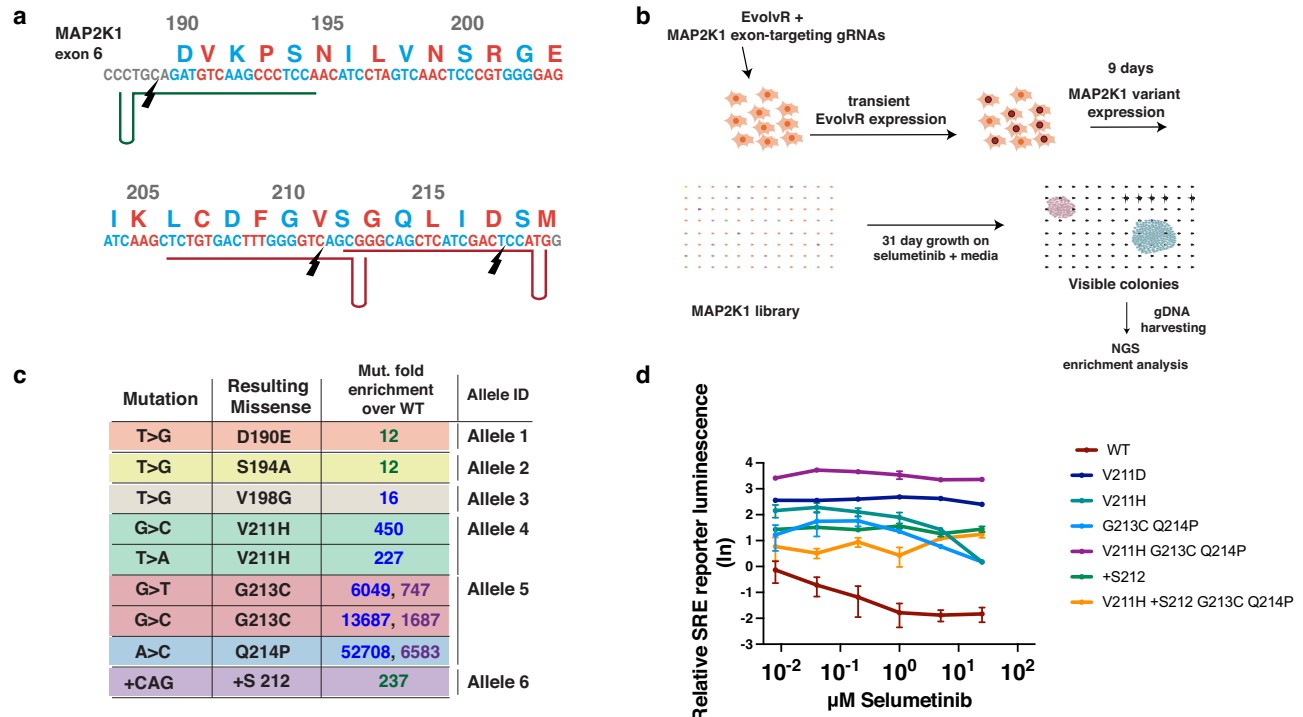

**Fig. 2 | EvolvR-mediated directed evolution uncovered drug-resistant MAP2K1 exon 6 variants. a** The diagram depicts the use of three gRNAs targeting the sense strand in green and the antisense strand in red of exon 6 of *MAP2K1*. Black markers represent enCas9 cut sites. **b** The diagram depicts the workflow used to evolve selumetinib-resistant *MAP2K1* variants using EvolvR. A375 cells are transiently transfected with plasmids encoding expression of EvolvR and gRNAs targeting *MAP2K1*. Cells were cultured without selection to allow *MAP2K1* variants to evolve before initiating selection with media containing 1 µM selumetinib. Cells were serially passaged until colonies growing rapidly under selective media were visible with the naked eye. gDNA was harvested from the cells and amplicon-sequenced to identify substitution mutations that are enriched in the population. **c** The table shows the substitution types and resulting missense mutations of variants enriched under selection of A375 cells by selumetinib after being transfected by EvolvR. All

substitution variants enriched cannot be generated by deaminases. Fold-enrichment is quantified by dividing the frequency of substitution variants generated by EvolvR by the frequency of those substitution variants in non-transfected cells. Allele labels indicate that mutations were co-represented and appeared together in all reads, resulting in a single selumetinib-resistant variant. Multiple numbers under "Mut. Fold enrichment over WT" correspond to the variant's enrichment in multiple replicates. Enrichment of variants in different biological replicates are distinguishable by the green, blue, and purple text in the column indicating fold-enrichment. **d** All variants enriched by at least 227-fold exhibited higher mean MAP2K1-ERK signaling activity than wildtype MAP2K1 at all concentrations tested as measured by the natural log of bioluminescence produced by SRE-driven expression of a luciferase reporter. Error bars show standard error of the mean for three biological replicates.

substitution in the V211H variant suggests that EvolvR generates all 12 substitutions at frequencies required for directed evolution, even if they appear below the limit of detection of Illumina sequencing in unselected libraries. Finally, we observed that one of the seven variants generated by EvolvR was an insertion variant resulting from an in-frame insertion of an additional serine in between V211 and S212 (+S212), demonstrating the potential value of EvolvR's capacity to generate functional indel variants in addition to substitutions.

## Nickase mismatch tolerance dictates the diversity of EvolvR-mediated substitutions within the gRNA target region

Though the known resistance-conferring T to A substitution in V211 was present and enriched by our screen, this substitution occurred only in combination with an additional G to C substitution in the codon encoding V211, resulting in a V211H missense mutation rather than the V211D mutation previously reported to confer resistance to selumetinib[34,35]. To explain the absence of the single substitution variant V211D, we first considered that SpCas9 frequently tolerates single mismatches within the 20 bp-long target region in a sequence-specific fashion, both within the seed region and PAM-distal nucleotides[36]. Though the enCas9 that we used for *MAP2K1* evolution is derived from an engineered enhanced Cas9 variant (eSpCas9.1.1)[37] that exhibits lower mismatch tolerance (due to its impaired sampling of the docked HNH conformational state necessary for DNA cleavage)[38], its improved fidelity is largely limited to

PAM-distal nucleotides and fails to reliably prevent the cutting of mismatched targets[39]. Understanding that enCas9 frequently tolerates mismatches throughout the target strand led us to hypothesize that mutation-containing target strands are re-nicked and subsequently overwritten by PolI5M using the wildtype sequence as a template to "erase" any previously introduced mutations. In contrast, mutations that sufficiently disrupt R loop formation and subsequent nicking by enCas9 are left to be processed by mismatch repair or "locked in" upon DNA replication. We henceforth refer to this mechanism as the "mismatch tolerance bias" (MTB) model for EvolvR's positional substitution biases (Supplementary Fig. 5). We note that the MTB model explains enCas9-PolI5M's apparently higher single substitution rates than wildtype nCas9-PolI5M both in the fluctuation assays originally described in *E. coli*[13] and in the BFP to GFP assay in HEK293 cells (Fig. 1d), especially as both assays measure EvolvR's substitution rates in PAM-distal nucleotides, where eSpCas9.1.1's nicking fidelity is highest relative to wildtype SpCas9[39].

Cas9's fidelity has previously been improved by using truncated gRNAs, which has been shown to decrease Cas9's off-target activity without decreasing its on-target efficiency for certain gRNAs[40]. Moreover, prior work in *E. coli* has demonstrated an improvement in the efficiency of single-nucleotide genomic edits using Cas9-induced homology-directed repair (HDR) when using 5'-truncated 18 nt gRNAs[41]. To test whether the efficiency of EvolvR mutagenesis can also be improved by supplying nCas9 with truncated gRNAs, we tested an

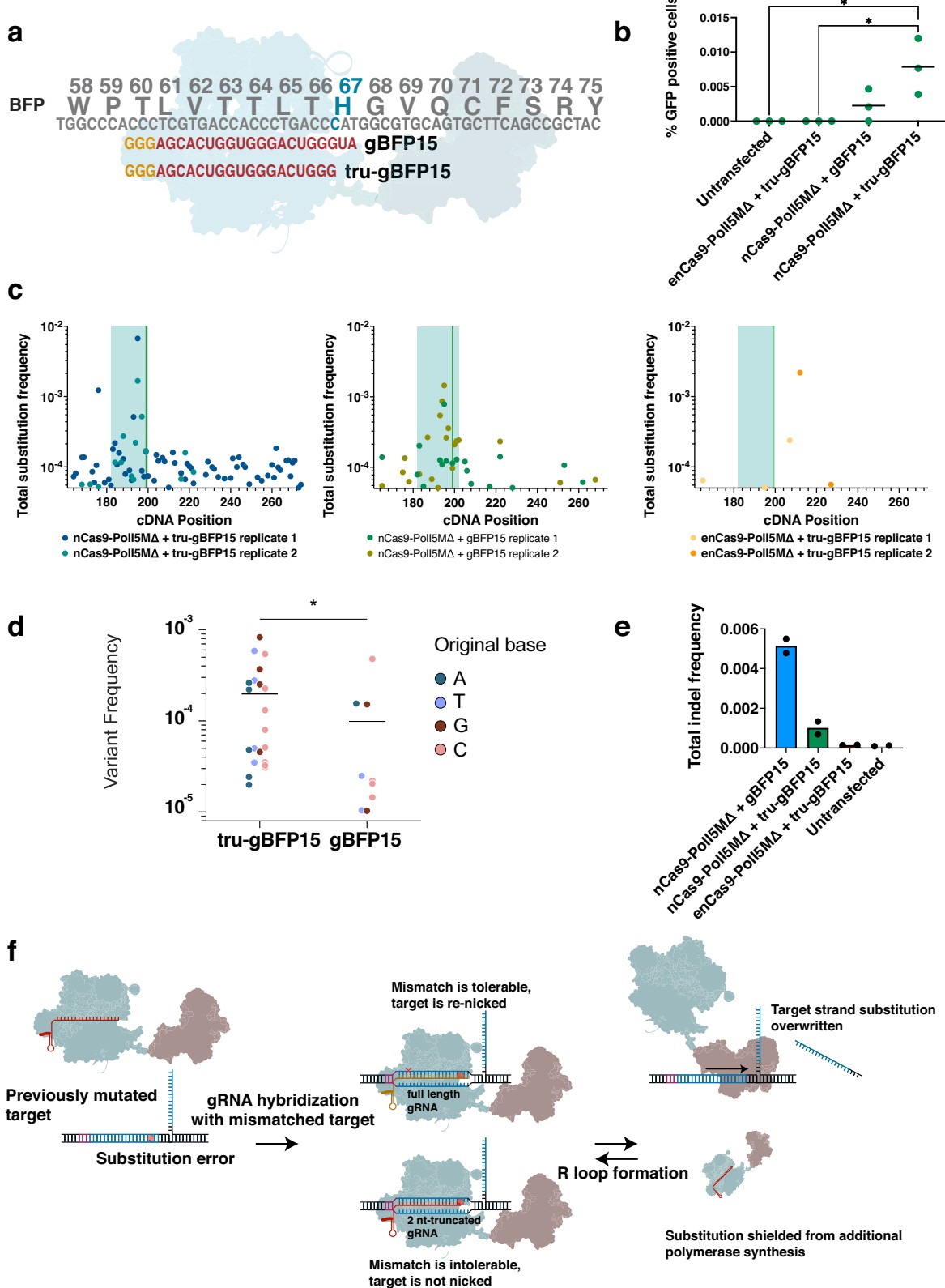

18 nt-long version of gBFP15 truncated by 2 nt at the 5' end and assessed nCas9-PolI5MΔ's activity with the BFP to GFP assay (Fig. 3a). The truncated guide elevated the frequency of GFP positive cells generated by nCas9-PolI5MΔ by 3.5-fold (Fig. 3b). In contrast, enCas9-PolI5MΔ showed no evidence of mutagenesis when directed by a truncated gRNA, presumably due to eSpCas9(1.1)'s sensitivity to the loss of PAM-distal gRNA-DNA base pairing[38].

As predicted by the MTB model, amplicon sequencing of the target locus showed a 4.7-fold increase in the mean frequency of substitution variants within the gRNA target region (Fig. 3c) and a 2.3-fold increase in the number of unique variants within 18 bp from the PAM site (Fig. 3d) when nCas9-PolI5M was coexpressed with a truncated 18 nt-long gRNA relative to with a full-length 20 nt-long gRNA. In addition to increasing the total frequency of substitution variants, the

**Fig. 3 | Enhanced nickase fidelity improves the diversity and frequency of EvolvR substitutions. a** Schematic represents two gRNAs of different lengths targeting the same target sequence within BFP, where one gRNA is 20 nt in length and the other 18 nt in length. gRNA sequences are labeled in red. gBFP15's PAM sequence is labelled in gold. **b** Truncating gBFP15 enhances EvolvR's mutation rate when nicking Cas9 is fused to PolI but not when a high-fidelity nickase is fused to PolI. The percentage of GFP positive cells generated by each construct was measured by flow cytometry and normalized by transfection efficiency. Green dots represent three separate biological replicates. Black lines represent mean percent GFP positive cells. Asterisks represent identical *p* values of 0.0144 as calculated by a one-way ANOVA followed by Tukey's HSD test. EvolvR cartoon is adapted from Halperin et al. 2018[13]. **c, d** The use of a truncated gRNA enhances the diversity generated by nCas9-PolI5M. The total frequency of substitution variants was quantified at each position within the BFP reference sequence (**c**). Positional coordinates are in reference to the BFP transgene's forward cDNA sequence in the 5′ to 3′ direction. The blue rectangle marks the gRNA target sequence region. The number of detectable unique substitution mutations for all four nucleotides increased when EvolvR was coexpressed with a truncated gRNA (tru-gBFP15) (**d**). Dots represent the mean frequency of individual substitution variants within the gRNA target sequence 18 bp from the PAM site in two biological replicates. Asterisk represents a *p* value of 0.0176 as calculated using a two-tailed student's t test. **e** The use of a truncated gRNA decreases the mean frequency of indels generated by nCas9-PolI5M. Dots represent the frequency of indels measured by amplicon sequencing of two biological replicates. **f** Diagram depicts proposed mechanism underlying increases in substitution rates at certain positions within the gRNA target sequence when tru-gBFP15 is used. Substitutions that are tolerable to nCas9 with full-length gRNAs may not be tolerable when complexed with truncated gRNAs, protecting new substitution errors on the target strand from additional cycles of DNA synthesis. EvolvR cartoon is adapted from Halperin et al.[13].

use of a truncated gRNA also decreased the frequency of indel variants by 5.1-fold compared to cells treated with full-length gRNAs (Fig. 3e). This suggests that reducing the number of tolerable substitution variants facilitates the installation of substitution variants before the appearance of indels, which are known to be far more likely to be intolerable to Cas9 nuclease activity than substitutions[39,42,43]. Additionally, as EvolvR can only be intolerant of substitutions that appear within the target sequence of its gRNA, its mutation rate should be markedly higher within the spacer region, consistent with the data shown here (Fig. 3c) and in prior work[13,27]. Together, these data show that EvolvR's biases are heavily influenced by target sequence-specific and nickase-specific mismatch tolerance, and that current measurements of EvolvR's substitution biases are not reflective of polymerase error rates alone.

## Incorporation of a PAM-flexible nickase with high activity and specificity enables consistent mutagenesis of a target nucleotide with most overlapping gRNAs targeting the sense strand

The mismatch tolerance model relates the diversity of EvolvR's edits to its ability to resist nicking mismatched targets. However, mismatch tolerance is highly gRNA-specific and depends on factors like the propensity of a given R-loop to form noncanonical base pairs[44], base-skipping[44], and gRNA-DNA free binding energy[45]. Moreover, truncation of gRNAs can impair R-loop formation[46], which is the rate-limiting step for Cas9 nuclease activity[47]. Accordingly, the use of truncated gRNAs alone is unreliable for improving the utility of EvolvR. In contrast, identifying or engineering RNA-guided nickases with reliably low mismatch tolerance and high activity when complexed with all or most gRNAs is an attractive engineering goal. Prior work has thus far only incorporated unengineered *S. pyogenes* nCas9 (D10A) and enhanced nicking Cas9 into EvolvR. However, other Cas9 variants with enhanced fidelities have been engineered, each of which could potentially endow EvolvR with more favorable on and off-target mutagenesis profiles. Therefore, we assessed numerous engineered Cas9 variants using the BFP to GFP mutagenesis assay.

A panel of Cas9 nickases derived from various engineered SpCas9 variants[37,38,44,48–51] was examined for the capacity to generate GFP positive variants in all of three biological replicates (for consistency) using the BFP to GFP conversion assay with various gRNAs. To emulate variations in R loop formation dynamics, we tested three gRNA variants in each of two distinct spacer regions, where one gRNA generates a nick 3 bp away from 199 C > T (gBFP3) and the other generates a nick 15 bp away (gBFP15). For each of the two spacer regions, we tested a 20 nt-long gRNA, an 18 nt-long gRNA, and a 20 nt-long gRNA with a guanine attached to its 5′ end, which is conventionally added to enhance gRNA expression from the human U6 promoter and initiate transcription from the first nucleotide of the gRNA sequence[52]. As anticipated, the performance of all Cas9 variants tested was highly sensitive to both the spacer itself and the design of the gRNA used

(Supplementary Fig. 6a, b). While nCas9, enCas9, Sniper-nCas9, LZ3-nCas9, HSC1.2-nCas9, HF1-nCas9 and SpRY-nCas9 all generated GFP positive cells consistently in at least one of the three designs for gBFP15, only nCas9, Sniper-nCas9, and SpRY-nCas9 generated GFP positive cells in at least two of three replicates using gBFP3. Therefore, we conclude that EvolvR's capacity to diversify a particular nucleotide is largely dependent on nuclease properties and gRNA design.

To compensate for gRNA variability in generating specific substitutions, we considered additional nickases that were highly active and specific and that in addition were PAM-flexible to maximize the number of candidate gRNAs that overlap with a target locus. In particular, we considered two RNA-guided nucleases known to exhibit improved specificity compared to SpCas9[53]: Nme2Cas9[54] and SlugCas9[55]. To improve upon Nme2Cas9s somewhat restrictive NNNNCC PAM site, we tested a variant of Nme2Cas9 known as eNme2-C.NR (henceforth referred to as eNme2.NR) engineered by phage-assisted continuous evolution (PACE) to recognize NNNNCN PAM sites[56]. SlugCas9 natively recognizes NNGR PAM sites[53]. We transfected BFP HEK293 cells with plasmids encoding EvolvR with either eNme2.NR-nCas9 or Slug-nCas9 as the nickase, in addition to one of a diverse panel of gRNAs (Supplementary Figs. 7 a, b). We tested 20 and 18-nt gRNAs for Slug-nCas9 and 21 and 23-nt gRNAs for eNme2.NR-nCas9, as Nme2Cas9 is conventionally paired with 23 nt gRNAs for optimal editing efficiency[54,56]. Of the 44 gRNAs tested across 22 unique spacer sequences with eNme2.NR-nCas9, only four generated GFP positive cells in at least two out of three biological replicates (Supplementary Fig. 7a). The poor mutagenicity of eNme2.NR-nCas9-PolI5MΔ may be attributable to Nme2Cas9's relatively weak nuclease activity in mammalian cells[53], though the degree to which eNme2.NR-Cas9's activity compares to wildtype Nme2Cas9 has not been rigorously quantified.

Of 24 gRNAs tested across 12 unique spacer sequences with Slug-nCas9, four generated GFP positive cells in all three replicates (Supplementary Fig. 7b). Of these four gRNAs, three of them were 20 nt in length and represented three of five 20 nt gRNAs that both generated nicks upstream of 199 C and overlapped with the 199 C > T substitution. Through the MTB model, these data suggest that Slug-nCas9 is likely to exhibit adequately low mismatch tolerance throughout its spacer region in addition to sufficient on-target activity. This finding would be consistent with prior characterization of SlugCas9's activity using thousands of gRNAs, revealing that it may be the most specific and among the highest activity Cas9 nucleases reported to date[53].

Given Slug-nCas9's markedly superior reliability in generating BFP to GFP substitutions when incorporated into EvolvR, we more thoroughly characterized Slug-nCas9-PolI5MΔ's performance using a panel of gRNAs targeting 8 different spacers near C199 and with various lengths. Because 5′ terminal purines (A or G) offer superior control of gRNA transcriptional start sites and efficient transcription from the U6 promoter[52], we tested the mutagenicity of each gRNA that did not

natively possess a 5′ terminal purine with and without 5′ terminal guanines (Supplementary Fig. 8a, b). Of the 40 gRNAs tested, 17 generated GFP positive cells in all three biological replicates, all of which contained 199 C > T within the spacer and downstream of the nick. These 17 successful gRNAs consisted of one gRNA of ten that had 18 bp of complementarity to the target site, three of nine with 19 bp of complementary, six of nine with 20 bp of complementary, six of ten with 21 bp of complementarity (Supplementary Fig. 8c), and one of two with 22 bp of complementarity. Both 20 and 21 nt of complementarity enabled targeted mutagenesis with most gRNAs. Though 20 nt gRNAs more frequently generated GFP positive cells in three of three replicates than 21 nt gRNAs, three of nine 20 nt gRNAs tested failed to generate GFP positive cells in more than one of three replicates, while only two of ten 21 nt gRNAs failed to generate GFP positive more than one replicate.

Intriguingly, two gRNAs with identical target-complementary sequences differing only by the presence or absence of an additional mismatched 5′ terminal guanine enabled markedly different substitution frequencies, where the mismatched 5′ terminal guanine gRNAs generated GFP positive cells at a 6.4-fold higher frequency (Supplementary Fig. 8c). To investigate whether adding mismatched 5′ terminal purines to gRNAs with 20 bp of complementary to the target site could reliably enhance or hinder EvolvR's mutation rates, we compared EvolvR's mutation rates when gRNAs 21 nt or 20 nt with an additional mismatched 5′ terminal purine were used to target BFP and found that the 5′ terminal mismatch could either increase or decrease EvolvR's efficiency in a gRNA-specific fashion (Supplementary Fig. 9a, b). Together, these data reveal that the use of gRNAs with at least 20 bp of complementarity to the target sequence are suitable for mutagenesis with Slug-nCas9 EvolvR, though the efficiency of mutagenesis may be optimized by the addition of a 21$^{st}$ complementary nucleotide or a mismatched 5′ terminal purine. At this stage, however, additional general principles enabling effective gRNA design were unclear.

While Slug-nCas9 natively recognizes NNGR PAM sites[53], a variant of Slug-nCas9 recognizing NNG PAM sites was recently developed via PACE[57]. Given that the human genome's GC content within any given 20 kb window is at least 30%[58], NNG PAM site recognition would offer the capacity to diversify virtually any region of any gene of interest with EvolvR. Using a panel of 21-22 nt-long gRNAs (depending on whether a natural 5′ terminal guanine existed at the end of each gRNA, Supplementary Fig. 10a), we tested the reliability of NNG-Slug-nCas9-PolI5MΔ in generating BFP to GFP edits. Among the 24 gRNAs tested, ones that generated nicks upstream of C199 most efficiently generated GFP positive mutants (Supplementary Figs. 10b, Fig. 4a), highlighting EvolvR's directionally asymmetrical mutation windows consistent with Pol I initiating DNA synthesis from the nick generated by Cas9. 11 out of 18 gRNAs enabling nicking upstream of C199 relative to the target strand generated GFP positive cells in at least two out of three replicates, indicating a favorable probability of generating substitutions at a given position in the gRNA target locus using a randomly selected gRNA with 21 nt of complementarity in 6-well plate format. Amongst these 12 gRNAs, the average frequency of GFP positive cells generated varied by up to 39-fold (Supplementary Figs. 10b, Fig. 4a), prompting us to investigate gRNA-specific features that are predictive of efficiently generating GFP positive cells.

One such candidate feature is the free energy change of R loop formation ($\Delta G_B$), which has previously been shown to affect the efficiency with which SpCas9 generates indels within genomic loci[45,59]. $\Delta G_B$ is a function of the difference between the free energy change of gRNA-DNA hybridization ($\Delta G_H$), and the combined free energy change of target DNA unwinding ($\Delta G_O$) and gRNA folding ($\Delta G_U$), all multiplied by a PAM multiplier variable ($\delta_{PAM}$) to account for observed differences in editing efficiencies across different SpCas9 PAM sites ($\Delta G_B = \delta_{PAM} (\Delta G_O - \Delta G_H - \Delta G_U)$)[45]. To test for similar thermodynamic correlations with EvolvR mutagenicity, we adapted the CRISPRoff pipeline[59]

to calculate $\Delta G_B$ for the gRNAs we tested with NNG-Slug-nCas9, excluding gRNAs that do not overlap with C199 to control for mismatch tolerance effects and setting the $\delta_{PAM}$ to 1 given the absence of data informing PAM-specific modifiers for NNG SlugCas9. Intriguingly, we found a weak but statistically significant positive correlation between the frequency of GFP positive cells generated by individual biological replicates and -$\Delta G_B$ (Supplementary Fig. 10c). We next examined the relationship between percent GFP positive cells and the difference between $\Delta G_O$ and $\Delta G_H$, without considering $\Delta G_U$, in consideration of the possibility that RNA folding does not meaningfully affect EvolvR's substitution frequencies within the set of target sequences testable with the BFP to GFP assay. Strikingly, the difference in free energy changes between DNA unwinding and DNA-gRNA hybridization ($\Delta G_O - \Delta G_H$) exhibited a strong and statistically significant correlation with the frequency of GFP positive cells generated by those gRNAs (Fig. 4b), appearing to emulate a dose-response curve. We note that the appearance of an S curve is unsurprising, as gRNAs generating GFP positive cells at frequencies below the detection limit of our assays are invariably measured at mutation frequencies of "0" even though they may consistently generate GFP positive cells at frequencies too low to be measured with typical cell culture throughput. Meanwhile, the frequency of GFP positive cells should not simply linearly increase with mutation rates, as the presence of other variants would prevent a population of 100% GFP positive cells. Nevertheless, the frequency of GFP positive cells appeared to increase linearly with increasing ($\Delta G_O - \Delta G_H$) past ~29 kcal/mol within the set of gRNAs we tested here.

To validate the frequency of GFP positive cells as a proxy for substitution frequencies throughout EvolvR's mutation window, we selected three gRNAs to characterize with Illumina amplicon sequencing: one possessing a moderate free energy change that generates GFP positive cells efficiently (gBFP-14), one possessing a moderate free energy change that does not generate GFP positive cells (gBFP-1), and one possessing an unfavorable low free energy change that does not generate GFP positive cells (gBFP+14) (Fig. 4c). As reflected by their relative BFP to GFP editing efficiency, only gBFP-14 generated significantly elevated substitution rates in the 3′ direction of its cut site relative to the off target control (Fig. 4d, f). Notably, we observed both transversion and transition mutations for all four nucleotides when using gBFP-14 (Fig. 4e).

The capacity of NNG-Slug-nCas9 to generate productive nicks leading to substitutions as part of the EvolvR fusion protein represents a critical leap in EvolvR's utility as a genomic diversifier, effectively removing PAM availability as a limitation of the EvolvR technology's applicability. To maximize diversification within a locus of interest, as a general guideline we recommend the use of 20-21 nt gRNAs, depending on which has the greater difference in free energy change between position-weighted RNA-DNA hybridization and DNA-DNA melting, though the data presented here do not determine whether the correlation between EvolvR efficiency and free energy of R loop formation breaks down with gRNAs exhibiting lower or higher predicted free energy outside of the range tested here. We also generally recommend the addition of 5′ terminal guanines to all gRNAs expressed from the human U6 promoter, not only to maximize transcription, but also because the use of alternative 5′ terminal nucleotides is known to result in mixed populations of gRNAs[52] of varying lengths and starting nucleotides, which may reduce the accuracy of predictions that rely on known gRNA sequences.

## Discussion

In sum, we establish EvolvR as a single-molecule diversifier mutating all four nucleotides within genomic loci in mammalian cells and elucidate the role of nickase fidelity as a critical emergent property dictating EvolvR's positional substitution biases and mutation rates. Moreover, NNG-Slug-nCas9-PolI5MΔ enables redundant targeting of genes of

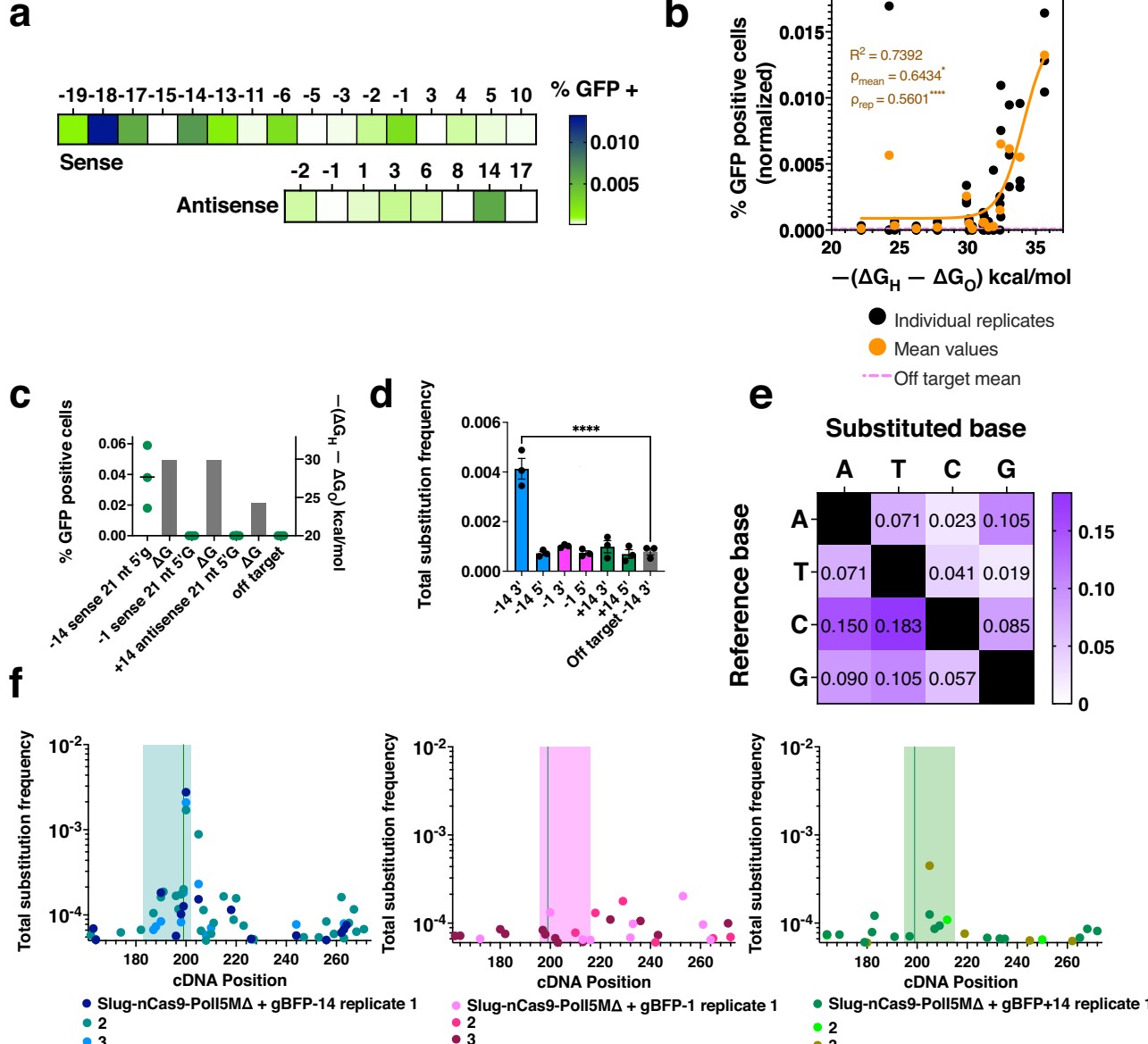

**Fig. 4 | High-fidelity, PAM-flexible nickase enables reliable mutagenesis of bp downstream of the nick and elucidates free energy as a determinant of gRNA-specific EvolvR efficiency. a** Heatmap cells show mean percent GFP positive cells normalized by transfection efficiency in three biological replicates. **b** The percentage of GFP positive cells generated by EvolvR is correlated with $-(\Delta G_H - \Delta G_O)$. $\rho_{rep}$ and $\rho_{mean}$ represent Spearman rank correlation coefficients for individual and mean values of percent GFP positive cells, respectively. Quadruple and single asterisks indicate $p$ values of less than 0.0001 and exactly 0.0278, respectively. The orange curve and R squared value correspond to a dose-response curve fit to percent GFP positive cells. **c** Green dots represent the percentage of GFP positive cells in three biological replicates. Black lines represent mean percent GFP positive cells. gRNA names correspond to gRNA labels in (**a**). Gray bars represent $-(\Delta G_H - \Delta G_O)$ in kcal/mol. **d** Black dots represent the total frequency of substitutions 40 bp

upstream and downstream of the gRNA-specific cut site in three biological replicates. Bar labels indicate the distance of each gRNA's cut site from C199 and the directionality of the sum of variants relative to the nicked strand. Bars of matching color show data for the same gRNA. Gray bar shows the frequency of substitutions 40 bp 3′ of position −14 for an off-target gRNA control. Quadruple asterisks represent a $p$ value of 0.0000015 as calculated by a one-way ANOVA and Tukey's HSD test. Error bars represent the standard error of the mean. **e** The proportion of substitutions generated by gBFP-14 of each type was normalized by the proportion of each variant's reference nucleotide within the target strand. **f** The frequency of substitutions was quantified at each position within the BFP reference sequence. Blue, pink, and green rectangles mark the gRNA target sequence region. The green vertical line marks the position of 199 C > T.

interest such that at least one available gRNA design is likely to efficiently diversify a given target codon despite variability in mutation rates observed across different gRNAs and gRNA-specific positional substitution biases. Finally, we leverage the broadened PAM flexibility of NNG-Slug-nCas9 to generate a large pool of gRNAs that can be tested with the sensitive BFP to GFP mutagenesis assay. These data were used to show that the free energy changes underlying R loop formation, which have already been used to predict gRNA-dependent

effects on Cas9 fidelity[45,59], are also predictive of EvolvR's mutation rates and may be used to computationally assess pools of gRNAs to avoid the need to screen them prior to directed evolution.

A limitation of EvolvR currently dictating its most practical applications in mammalian cell directed evolution is that its mutation window is limited by the number of gRNAs used to generate individual nicks. In this study, our fruitful use of EvolvR to identify drug-resistant *MAP2K1* variants with three gRNAs suggests that EvolvR is immediately

applicable for screening variants of short functional domains of interest (e.g. T cell costimulatory domains, antibody complementarity determining regions, etc.) or to more richly investigate sites where gain of function mammalian cell variants have been identified by base editors[60]. However, expanding EvolvR's mutation window to hundreds or thousands of bp to conduct more expansive surveys of functional landscapes like those enabled by processive base editors like HACE would currently require the ability to multiplex large numbers of gRNAs. Accordingly, the aim of increasing the mutation window per nick generated could motivate the engineering of DNA polymerases that are both highly processive and error-prone in order to decrease the number of gRNAs that need to be delivered to target cells simultaneously.

Collectively, our findings pave a path to generating a complete set of missense variants for any codon in the mammalian cell genome, facilitating the directed evolution of native genomic loci in mammalian cells via genetic diversification not previously attainable by in vivo diversifiers. Critically, we also reveal the nuclease component of EvolvR to be an attractive subject of engineering in addition to the polymerase, elucidating a means of improving the technology in future work.

## Methods

### Plasmid construction
Plasmids used in transfections were assembled using Golden Gate cloning. Plasmids used for transient transfection of 293 BFP and A375 cells contained a ColE1 origin of replication and AmpR resistance cassette. Prior to transfection, plasmids contained an sfGFP expression cassette in between human U6 promoter and a gRNA scaffold flanked by BsmBI cut sites that would generate overhangs 5′GGTG and 5′TTTG or 5′GTTG depending on the gRNA scaffold sequence. Oligonucleotides containing gRNA sequences and the appropriate overhang sequences 5′CACC and 5′AAAC or 5′CAAC were annealed and knocked in by Golden Gate reaction. A comprehensive table of the oligonucleotides used to generate gRNAs in this study is available in the Supplementary Data 1.

### Mammalian cell culture
HEK293T and A375 cells were obtained from the UC Berkeley Cell Culture Facility located in 336 Barker Hall, Berkeley, CA 94720. A375 cells were originally sourced from American Type Culture Collection (ATCC CRL-1619IG-2). BFP HEK293 cells were generously donated by Jacob Corn's lab.

Cells were cultured in Dulbecco's modified Eagle's medium (DMEM) plus L-glutamine (Gibco) supplemented with 10% (v/v) fetal bovine serum (Gibco, qualified) and 1x antibiotic-antimycotic. Cells were incubated at a temperature of 37 degrees Celsius and at 5% CO2 concentration.

### BFP to GFP mutagenesis experiment in BFP HEK293 cells
Unless otherwise noted, cells were transfected using Mirus X2's transfection reagent according to the manufacturer's protocol. Briefly, transfection complexes were prepared by mixing DNA in a 3:1 µg DNA to µL Mirus *Trans*IT-X2 (cat no. MIR 6004) reagent ratio in 250 µL Opti-MEM serum-free medium (cat no. 31985070). The mixture was incubated at room temperature for 15–30 minutes before dispensing on top of cells. Cells were harvested at 70–90% confluence and reverse transfected by seeding cells at 1 million cells/well in tissue-culture treated 12-well plates immediately before adding transfection mixtures.

16–24 h after transfection, cells were trypsinized and harvested from 12-well plates and resuspended in 1 mL PBS. 250 µL were transferred to a 96-well plate for flow cytometry analysis on an Attune Nxt flow cytometer, while the remaining 750 µL were separated into triplicate 6-well plate wells by dispensing 250 µL into each well.

120–144 h later, cells were once again harvested by trypsinization, resuspended in 300 µL PBS in 96-well plates, and analyzed on an Attune NxT or Sony SH800Z. Transfection rates and GFP frequencies were determined by analysis on FlowJo of the YL1 and BL1 channels collected on the Attune NxT or the mCherry and EGFP channels collected on the Sony SH800Z.

For high-throughput screens of various high-fidelity nickases shown in Supplementary Fig. 6, transfections were scaled down to 48-well plates by seeding 30,000 cells in 48-well plates per condition one day prior to transfection. When the cells were 70–90% confluent, the cells were transfected with Mirus *Trans*IT-X2 reagent according to the manufacturer's protocol by mixing 390 ng DNA in a 1:3 ng DNA to µL transfection reagent ratio with 1.17 µL transfection reagent in 39 µL Opti-MEM reduced serum medium. The cells were transfected and incubated as described previously. 16–24 h later, the cells were resuspended in 400 µL enzyme-free dissociation buffer (Gibco) and split into four 100 µL volumes, where three were passaged into 2 mL growth media in 6-well plates and the remaining 100 µL volume was analyzed for mCherry fluorescence on an Attune NxT flow cytometer. The cells were incubated at 5% $CO_2$ and 37 degrees Celsius for ~7 days to enable mutagenesis and were then harvested and analyzed by flow cytometry as previously described.

### Next-generation sequencing experiments with HEK293 BFP cells
HEK293 BFP cells were seeded at ~5,000,000 cells per plate in 10 cm tissue culture plates. When cells reached approximately 70% confluence, transfection complexes were prepared by mixing 15 µg plasmid with 45 µL Mirus *Trans*IT-LT1 (cat no. MIR 2304) reagent ratio in 1.5 mL Opti-MEM serum-free medium. The mixture was incubated at room temperature for 15-30 minutes before dispensing on top of cells. ~36 h later, cells were trypsinized and resuspended in 350 µL PBS and passed through a cell strainer into a FACS tube. 100,000 mCherry positive cells from each transfection were sorted on a Sony SH800Z sorter into 2 mL growth media in 15 mL Falcon tubes for each biological replicate, seeded onto 10 cm plates, and incubated for two weeks.

The cells were resuspended in 200 µL PBS and gDNA was harvested using a Qiagen DNEasy kit according to the manufacturer's protocol.

### Library preparation for next generation sequencing of BFP
The BFP locus was amplified from 500 ng of gDNA for 20 cycles of PCR using oligonucleotides GCTCTTCCGATCTNNNNNACCCTGAAGTT-CATCTGCACCA and GCTCTTCCGATCTNNNNNTTGAAGAA-GATGGTGCGCTCCT. The PCR product was purified using PCR Reaction Cleanup beads from the UC Berkeley Sequencing Facility located in 310 Barker Hall, Berkeley, CA 94720. The beads were warmed to room temperature and mixed thoroughly in 9:5 volumetric ratio with the PCR product and incubated at room temperature for 5 minutes. Beads were separated from the solution by placing them on magnetic racks and washed with 70% ethanol twice. Ethanol was removed and beads were allowed to dry at room temperature for 30 minutes. The beads were removed from the magnetic rack and thoroughly resuspended in water to elute PCR 1 product. The concentration of PCR 1 product was quantified with a Qubit fluorometer using the high-sensitivity DNA kit. 10 ng of PCR 1 product was added to 10 additional cycles of PCR, in which Illumina adapter sequences and unique dual index pairs were attached. The samples were delivered to the University of California Berkeley Vincent J. Coates Genomics Sequencing Laboratory, where the samples were purified via bead cleanup and size selected by Pippin Prep. The molar concentration of each library prep was quantified by qPCR prior to pooling. The library was sequenced using an Illumina MiSeq V2 150 paired-end kit on an Illumina MiSeq sequencer. Samples were demultiplexed using bcl2fastq (v2.20).

## Sequencing data analysis and variant calling

The first 5 nucleotides of each read were trimmed to eliminate low quality reads using seqtk. Trimmed fastq.gz files were merged using NGmerge (version 0.3) and were filtered for any read pairs that were not perfectly matching with "-p 0". The merged fastq files were aligned with the BFP reference sequence with bowtie2 (version 2.4.4). The resulting BAM files were indexed and converted to SAM files using samtools (version 1.13). Mpileup files were generated from the resulting SAM files using samtools mpileup, filtering for bases calls with a quality score of at least 30 using -Q 30 and no mapping quality filter. Variant calling was performed by running mpileup files through SiNPle (version 0.5)[61] using a theta value (prior probability of a substitution variant) of 0.9.

To reduce the influence of base calling errors and prior mutations on variant calling, the mean variant frequency at each position for 2-3 untransfected negative control replicates was subtracted from the frequency of that same variant in all other conditions for all possible variants. Then, substitutions appearing at less than 0.00001 frequency were set to 0 to filter out variants that do not appear in at least 1/100,000 reads.

To calculate the relative proportion of each substitution type within 40 bp of the nick within the total set of substitution variants detected, the number of unique variants of each type of substitution detectable in at least 1/100,000 reads after background subtraction was divided by the total number of unique substitution variants detected above 1/100,000 reads after background subtraction. The relative proportions of each substitution type were then normalized by the relative proportion of the reference nucleotide within 40 bp of the nick to account for the differences in abundance of each nucleotide in the original target sequence.

## MAP2K1 evolution

A375 cells were transfected using Mirus TransIT-X2's transfection reagent according to the manufacturer's protocol. Briefly, cells were seeded in 6-well plates one day prior to transfection. Once they reached 70% confluence, transfection complexes were prepared by mixing 1 μg plasmid in 3 μL Mirus TransIT-X2 reagent in 250 μL Opti-MEM serum-free medium. The mixture was incubated at room temperature for 15-30 minutes before dispensing on top of cells.

Approximately 60 h later, the cells were trypsinized and split in a 1:4 fashion into separate 10 cm tissue culture plates, and the remaining cells were analyzed on a Sony SH800Z cell sorter to confirm the efficient transfection of each sample. After one week of growth, the cells were split at a 1:10 ratio into new 10 cm dishes with selective growth media containing 1 μM selumetinib. Selective media was replaced routinely every 48–72 h until large colonies were visible by the naked eye (~41 days later).

The cells remaining on each plate were harvested and gDNA was harvested with a Qiagen DNEasy Blood & Tissue Kit (cat no. 69045) according to the manufacturer's protocol.

## Library preparation for MAP2K1 evolution experiment

A section of the *MAP2K1* exon 6 locus was amplified from 360 ng of gDNA for 20 cycles of PCR using forward primer GCTCTTCCGATCT NNNNNCCCTCCTTTTCTATTTTCTCTTCCCTGCAG and reverse primer GCTCTTCCGATCTNNNNNCCGACATGTAGGACCTTGTGCCC. The PCR product was purified using PCR Reaction Cleanup beads from the UC Berkeley Sequencing Facility located in 310 Barker Hall, Berkeley, CA 94720. The beads were warmed to room temperature and mixed thoroughly in 9:5 volumetric ratio with the PCR product and incubated at room temperature for 5 minutes. Beads were separated from the solution by placing them on magnetic racks and washed with 70% ethanol twice. Ethanol was removed and beads were allowed to dry at room temperature for 30 minutes. The beads were removed from the magnetic rack and thoroughly resuspended in water to elute PCR 1

product. The concentration of PCR 1 product was quantified with a Qubit fluorometer using the high-sensitivity DNA kit. 10 ng of PCR 1 product was applied to a second PCR, in which Illumina adapter sequences and unique dual index pairs were attached. The samples were delivered to the Innovative Genomic Institute Sequencing Core, where the samples the molar concentration of each library prep was quantified by qPCR prior to equimolar pooling. The library was sequenced using an Illumina MiSeq V2 150 paired-end kit on an Illumina MiSeq sequencer. Samples were demultiplexed using bcl2fastq (v2.20).

## Sequencing analysis for MAP2K1 evolution experiment

Raw fastq.gz files were processed into SiNPLe variant calling files as described in "Sequencing data analysis and variant calling". To calculate fold-enrichment of each substitution, the frequency of each variant in EvolvR on target conditions were divided by the frequency of those variants in the untransfected controls not having undergone selection.

Allele tables were generated using CRISPRESSO version 1.0.13.

## SRE reporter assay

A plasmid DNA CMV-driven expression of human *MAP2K1* was obtained from Origene (cat no. RC218460). Variants of interest were subsequently cloned by site directed mutagenesis using PCR with substitution-containing primers (IDT). Expression cassettes for the variants of interest were assembled via Golden Gate assembly.

HEK293 cells were seeded at 30,000 cells per well in 96 well white assay plates. After overnight incubation, the cells were transfected with Lipofectamine 2000 according to the manufacturer's protocol with 50 ng pMAP2K1. -7 h later, the cells were washed twice with PBS and incubated with media containing selumetinib for 12 h. After 12 h of incubation at 37 degrees Celsius and 5% $CO_2$, the cells were washed with twice PBS and their media was replaced with media containing 10 ng/mL human epidermal growth factor (hEGF) and 0.5% FBS. -6 h later, MAP2K1 activity was measured using the SRE Reporter Kit (BPS Bioscience cat no. 60511) according to the manufacturer's protocol using a Tecan Spark plate reader. MAP2K1 activity was quantified according to the manufacturer's protocol. Briefly, mean background luminescence from cell-free wells was subtracted from the luminescence from all other wells. To determine relative MAP2K1 activity between wells, background-subtracted Firefly luciferase luminescence was divided by luminescence generated by Renilla luciferase, a luminescent reporter of transfection efficiency.

## Calculation of free energies

The free energies of DNA-DNA separation $\Delta G_O$ and DNA-RNA hybridization $\Delta G_H$ were calculated using the calcDNAopeningScore and calcRNADNAenergy functions, respectively found in the publicly available CRISPRoff pipeline v1.1.2 (https://github.com/rth-tools/crisproff/). The output of calcRNADNAenergy was weighted by position as in the get_eng function when pos_weight = True. The free energy of RNA folding was calculated using the ViennaRNA package (RNAfold v2.7.0).

## Statistics & reproducibility

All figures present data as mean values of biological replicates. Where error bars are shown, they represent the s.e.m. Sample size and the statistical tests used for each experiment are two tailed student's t test for single comparisons and ANOVA followed by Tukey's HSD for multiple comparisons. No statistical methods were used to predetermine sample size. Statistical analysis was performed using GraphPad Prism (v10.4.0). No statistical method was used to predetermine sample size. Experiments were not randomized and investigators were not blinded to allocation during experiments and outcome assessment. No data were excluded from the analyzes.

## Reporting summary

Further information on research design is available in the Nature Portfolio Reporting Summary linked to this article.

## Data availability

Source data for all figures in this study are provided in the Source Data file. Next generation sequencing data used in this study are available in the NCBI SRA database under accession code PRJNA1157996. Source data are provided with this paper.

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

## Acknowledgements

We thank Flo Ramirez at the University of California, Berkeley Vincent J. Coates Genomics Sequencing Laboratory and Netravashi Krishnappa for assistance with high-throughput sequencing, Jacob Corn's lab for generously gifting us the BFP-expressing HEK293 cells used in this work, Libby H. Koolik for assistance with next-generation sequencing data analysis, and Christopher Allen for providing feedback on the manuscript. This research was supported in part by NIH T32GM139780 (to J.E.H.) the Laboratory for Genomics Research (LGR) Innovation Award (to J.E.H., J.E.D., and D.V.S.), Siebel Scholars (to J.E.H.), and the CRISPR Cures for Cancer Initiative (to D.A., J.E.D., and D.V.S.).

## Author contributions

J.E.H. conceived the study, designed and executed all experiments, analyzed and interpreted all of the data, and wrote the manuscript. A.J.S. and S.O.H. contributed to experimental design. J.G. contributed to plasmid construction and execution of the BFP to GFP mutagenesis assay. A.J.S., S.O.H., J.G., D.A., J.E.D., and D.V.S. contributed to interpretation of data and provided critical feedback on the manuscript. J.E.D. and D.V.S. supervised the study.

## Competing interests

The Regents of the University of California have submitted a patent application pertaining to the use of truncated gRNAs for enhancing EvolvR's mutation rates to the World Intellectual Property Organization under the PCT system (application number PCT/US2023/069704). J.E.H., A.J.S., J.E.D., and D.V.S. are listed as inventors. The Regents of the University of California have submitted a provisional patent application to the United States Patent and Trademark Office (application number 67/747,791) pertaining to the use of EvolvR to diversify genomic loci in mammalian cells. J.E.H., A.J.S., J.E.D., and D.V.S. are listed as inventors. The remaining authors declare no competing interests.
