## [Transparent Peer Review file · Nature Communications]

Nickase fidelity drives EvolvR-mediated diversification in mammalian cells

Corresponding Author: Professor John Dueber

Version 0:

Reviewer comments:

Reviewer #1

(Remarks to the Author)
Summary

Hurtado et al. report an EvolvR system suitable for targeted mutagenesis in mammalian cells. Using a Cas9 nickase fused to a mutagenic DNA polymerase, the authors demonstrate diversification of ~50 bp of genomic DNA yielding a red-shifted blue fluorescent protein and novel selumetinib-resistant MAP2K1 variants. Finally, incorporation of a PAM-flexible Cas9 nickase is shown to expand the possible targeting range of the construct.

Significance

This study extends the host range of EvolvR beyond *E. coli* and *S. Cerevisiae* (previously reported) to enable targeted mutagenesis in mammalian cells. While the ~50 bp mutagenesis window of EvolvR is modest (for example, compared to the multi kilobase windows of T7-polymerase guided methods), generation of transversion mutations is quite important because it can unlock access to the full complement of missense mutations required for unrestricted amino acid conversions. Though the mutagenesis strategy remains identical to previous iterations of EvolvR, in mammalian cells use of a PAM-flexible and highly specific nickase significantly expanded potential targeting range and reduced mutation reversion due to re-targeting of edited loci. The requirement for extensive sgRNA optimization/screening and lack of clear design rules may make implementation of EvolvR cumbersome in practice.

Overall Assessment

EvolvR should prove to be a useful addition to extant methods for targeted diversification in mammalian cells. The limited editing region and likely large amounts of optimization needed to effectively employ EvolvR as a genetic diversifier may somewhat limit the method's utility. While these results do not significantly advance the conceptual paradigm for the field of targeted in vivo mutagenesis, as EvolvR here functions essentially the same as EvolvR in other systems, the engineering efforts and validation provided in this manuscript are substantial and unlikely to be included in a paper about evolving an interesting target (rather than the proof-of-concept applications here), so the paper has substantive merit.

Major Concerns

1. A complete characterization of the mutational spectrum and mutation rate is required. While the EvolvR system is touted as generating all 12 substitution mutations, only a C→T transition (BFP→GFP experiment) and four types of transversions (MAP2K1 experiment) are demonstrated. Demonstration and discussion of the PolI5MΔ mutational spectrum in mammalian cells is essential.
2. A key issue is the lack of evaluation of the off-target editing of EvolvR in mammalian cells. Targeted mutagenesis is a critical aspect of EvolvR and, without it, the use case for the system precipitously drops. The data in Figure 1e are noteworthy in this regard and greatly exacerbate this potential issue, as enCas9-PolI5mΔ supplied with off-target gRNA resulted in noticeably elevated mutation in the non-targeted, amplified region. The authors need to experimentally address and thoroughly discuss this issue.

3. The sgRNA dependence problem is highlighted by the PAM-flexible section (line 313). How will effective editing of endogenous loci be practically assessed, so that useful guides can be identified rapidly? sgRNA screening will be more complicated as there is not a fluorescent reporter for endogenous loci.
i. The number and types of sgRNA modifications (e.g., spacer length, 5'-G terminal capping) require significant screening and optimization for EvolvR to work robustly, making practical implementation of the method rather burdensome. No clear sgRNA design principles were gleaned from Extended data Fig 5-7. It would help a lot if clear design principles were obtained and clearly provided.

Minor Comments

-While the PAM-flexible nickases are certainly an engineering improvement, amplicon sequencing of the target region should be performed with the PAM-flexible nickases before any conclusions are drawn about the mutational efficacy of these chimeras compared to enCas9. A direct comparison of the mutation rate and spectrum is highly recommended.

-The introduction should include a discussion of the TRACE targeted mutagenesis system (with a direct reference), especially with respect to the following point.

-The limited targeting region of EvolvR limits its applicability for evolving large genes or metabolic pathways – applications suggested in the conclusions that appear unlikely. Practical use of EvolvR necessitates prior knowledge about where to direct the editor, making it less useful for exploratory evolution experiments in native genomic contexts. Most particularly, evolving large genes or gene clusters (e.g., metabolic networks as posed in the conclusion) will be challenging since sgRNA tiling will be required.

-Lines 92–94 suggest the adenoviral evolution system has the same cheating problem as VEGAS. As far as I know, that is not the case.

-Introducing Map2K1 with 1–2 sentences would help readers understand why it is important etc.

-A discussion of CRISPR NHEJ methods that have been shown to introduce transversions in vivo in mammalian cells at endogenous loci could help clarify the practical advance over existing technologies.

-DNA sequences should be included in the Supplemental Information.

-Depositing the plasmids in Addgene would significantly amplify the accessibility of this research.

-Lines 357-8 need a reference for PAM flexible Slug-Cas9

-Flow cytometry gating examples and raw data should be provided as supplementary information.

Reviewer #2

(Remarks to the Author)

CRISPR-guided DNA polymerases enable diversification of all nucleotides in a tunable window in E.coli, the authors of this paper tested whether EvolvR can diversify targeted genomic loci in human cells. They demonstrated EvolvR's ability to generate transition & transversion mutations across ≥ 50 bp, evolving novel drug-resistant MAP2K1 variants via substitutions unattainable by deaminases. Moreover, they found that EvolvR's mutation window and substitution biases are influenced by the gRNA design and nickase properties, they incorporated a PAM-flexible nickase into EvolvR to make an efficient genetic diversifier. Nonetheless, I expect the authors to conduct a few more additional characterizations to improve the manuscript for eventual publication in Nature Communications.

Major Suggestions:

1. It has been reported in published literature that EvolvR mainly mutates the A and T bases in the target area in E.coli. I think it is necessary to describe the base preference of EvolvR in mammalian cells in detail.
2. In Figure 3, the authors used 18 nt-long gRNAs truncated by 2 nt at their 5' end and assessed enCas9-PolI5M activity according to prior work in E. coli, while in mammalian cells, could more sgRNAs with different numbers of truncations not only 2 nt to be tested, to further enhance the mutation efficiency of EvolvR?
3. In Figure 3e, the authors found truncated gRNA reduces nCas9-PolI5M-induced indel frequency vs. full-length gRNA, However, only two repeated experiments are not enough to support the conclusion, and the number of repeated experiments needs to be increased.
4. Whether the off-target effects of the CRISPR component in EvolvR potentially elevate the incidence of false positive results during mutation screening?

Minor Suggestions:

5. The author should describe the window size of mutations generated by EvolvR, and what is the maximum range?
6. The resolution of all figures needs to be improved.
7. In conclusion part, the limitations of EvolvR and possible solutions should be added.

Reviewer #3

(Remarks to the Author)

Reviewer #4

(Remarks to the Author)

Version 1:

Reviewer comments:

Reviewer #1

(Remarks to the Author)

The authors have effectively addressed my comments. I congratulate them on a very nice paper.

Reviewer #2

(Remarks to the Author)

The authors have addressed all the concerns from this reviewer.

Reviewer #3

(Remarks to the Author)

Reviewer #4

(Remarks to the Author)

We thank the reviewers for their insightful and constructive feedback, which we have applied to significantly improve the quality of the attached revised manuscript as detailed below.

REVIEWER COMMENTS

Reviewer #1 (Remarks to the Author):

Summary

Hurtado et al. report an EvolvR system suitable for targeted mutagenesis in mammalian cells. Using a Cas9 nickase fused to a mutagenic DNA polymerase, the authors demonstrate diversification of ~50 bp of genomic DNA yielding a red-shifted blue fluorescent protein and novel selumetinib-resistant MAP2K1 variants. Finally, incorporation of a PAM-flexible Cas9 nickase is shown to expand the possible targeting range of the construct.

Significance

This study extends the host range of EvolvR beyond *E. coli* and *S. Cerevisiae* (previously reported) to enable targeted mutagenesis in mammalian cells. While the ~50 bp mutagenesis window of EvolvR is modest (for example, compared to the multi kilobase windows of T7-polymerase guided methods), generation of transversion mutations is quite important because it can unlock access to the full complement of missense mutations required for unrestricted amino acid conversions. Though the mutagenesis strategy remains identical to previous iterations of EvolvR, in mammalian cells use of a PAM-flexible and highly specific nickase significantly expanded potential targeting range and reduced mutation reversion due to re-targeting of edited loci.

We thank the reviewer for this positive feedback.

The requirement for extensive sgRNA optimization/screening and lack of clear design rules may make implementation of EvolvR cumbersome in practice.

We have addressed this concern as described below in individual comments.

Overall Assessment

EvolvR should prove to be a useful addition to extant methods for targeted diversification in mammalian cells. The limited editing region and likely large amounts of optimization needed to effectively employ EvolvR as a genetic diversifier may somewhat limit the method's utility. While these results do not significantly advance the conceptual paradigm for the field of targeted in vivo mutagenesis, as EvolvR here functions essentially the same as EvolvR in other systems, the engineering efforts and validation provided in this manuscript are substantial and unlikely to be included in a paper about evolving an interesting target (rather than the proof-of-concept applications here), so the paper has substantive merit.

We thank the reviewer for this positive feedback and agree, especially with additions to this revised manuscript, that this manuscript presents innovations in the system design as well as novel design principles that are generally useful to those harnessing EvolvR.

Major Concerns

1. A complete characterization of the mutational spectrum and mutation rate is required. While the EvolvR system is touted as generating all 12 substitution mutations, only a C→T transition (BFP→GFP experiment) and four types of transversions (MAP2K1 experiment) are demonstrated. Demonstration and discussion of the PolI5MΔ mutational spectrum in mammalian cells is essential.

We appreciate the reviewer's request for a more detailed characterization of PolI5MΔ's "mutational spectrum". We now include the requested characterization for enCas9-PolI5MΔ and NNG Slug-nCas9-PolI5MΔ (Figures 1g, 4f), but do emphasize that these data should be understood alongside the following qualifiers: because the specific substitutions that are installed within EvolvR's substitution window are dictated by (1) the relative frequency of each nucleotide in the target sequence, (2) the biochemical properties of the nickase being used (Supplementary Fig. 6), and (3) gRNA-specific effects (Figure 3c,d), quantifying EvolvR's mutational spectrum using a given gRNA does not provide a broadly representative survey of its substitution biases. Critically, using the gRNAs described in the manuscript, we find that 11 (Figure 1g) or 12 (Figure 4f) of the possible 12 substitution variants are detected. Moreover, although A to C and T to G substitutions were consistently the most rare when using either enCas9 or Slug-nCas9, their presence within our hits from the MAP2K1 screen indicates that PolI5M can generate all 12 substitutions at sufficient frequencies to generate hits, even if they are the substitutions least frequently generated by EvolvR.

2. A key issue is the lack of evaluation of the off-target editing of EvolvR in mammalian cells. Targeted mutagenesis is a critical aspect of EvolvR and, without it, the use case for the system precipitously drops. The data in Figure 1e are noteworthy in this regard and greatly exacerbate this potential issue, as enCas9-PolI5MΔ supplied with off-target gRNA resulted in noticeably elevated mutation in the non-targeted, amplified region. The authors need to experimentally address and thoroughly discuss this issue.

We agree with Reviewer 1 that targeted mutagenesis is an important feature of EvolvR but differ in our assessment of the data presented in our manuscript as outlined below.

- i. As Reviewer 1 correctly points out, enCas9-PolI5MΔ supplied with an off-target gRNA appears to result in elevated substitution frequencies within some positions in at least one of two replicates in Figure 1f (previously Figure 1e). However, they consist of only 6 positions 50 bp downstream of the nick generated by the on-target position. These data, in addition to the absence of GFP positive cells detected via the BFP to GFP mutation experiment (Figure 1d), reveal that EvolvR is highly unlikely to mutate a given nucleotide residing outside the on-target

mutation window, and that the density of EvolvR off-target mutations is far lower outside of the on-target mutation window than within it.

- ii. For the directed evolution of cultured cells, off-target mutagenesis is detrimental when it is sufficiently elevated to generate false hits consisting of substitution variants outside of the gene of interest (though these hits may be serendipitously valuable as well). Moreover, the validation of hits identified in any directed evolution campaign is routinely performed in a clean genetic background to rule out fitness-enhancing off-target effects even in applications using genetic diversifiers with virtually no off-target mutagenesis (see Ravikumar et al. 2018¹ and Rix et al. 2020² using OrthoRep for example). To this end, our independent validation of highly enriched MAP2K1 variants using the SRE reporter assay (Figure 2d) demonstrates that hits generated by EvolvR were not enriched by virtue of being associated with an off-target mutation outside of MAP2K1.
- iii. EvolvR may generate off-target substitutions through two potential mechanisms: gRNA-independent mutagenesis arising from PolI5M initiating synthesis from pre-existing (i.e. Cas-independent) nicks, or gRNA-dependent mutagenesis arising from off-target cleavage by the nuclease. As off-target nicking is unlikely to occur at loci that confer functional diversity identical to that being selected for and should vary widely by gRNA, we were primarily concerned with the potential for gRNA-independent elevation of global mutagenesis. As detecting off-target mutations at frequencies at or below those generated by EvolvR in our experiments at on-target sites (10^{-5} - 10^{-3} substitutions/bp) would require a sequencing depth greater than that afforded by ultra-deep whole genome sequencing (e.g using a NovaSeq S4 yielding a maximum of 20 billion reads), we opted to more sensitively measure the incidence of mutating a particular bp within the human genome via gRNA-independent mechanisms using the frequency of BFP to GFP missense mutants generated by an off-target or inactive gRNA control (see Figures 1d and 3b for example). From these experiments, we conclude that PolI5M Δ does not detectably generate substitutions throughout the genome within a 6-well plate or 10 cm plate tissue culture.

In summary, we share Reviewer 1's view that EvolvR's capacity to elevate substitutions specifically within target loci is an essential part of the manuscript's story and believe that the data presented in the manuscript rule out off-target mutagenesis at frequencies that would significantly hinder its use for directed evolution.

3. The sgRNA dependence problem is highlighted by the PAM-flexible section (line 313). How will effective editing of endogenous loci be practically assessed, so that useful guides can be identified rapidly? sgRNA screening will be more complicated as there is not a fluorescent reporter for endogenous loci.

- i. The number and types of sgRNA modifications (e.g., spacer length, 5'-G terminal capping) require significant screening and optimization for EvolvR to work robustly, making practical implementation of the method rather burdensome. No clear sgRNA design principles were gleaned from Extended data Fig 5-7. It would help a lot if clear design principles were obtained and clearly provided.

We appreciate Reviewer 1's compelling communication of this important practical consideration and agree that the elucidation of design principles for gRNAs given the variation in performance between gRNAs would be highly enabling. Motivated by this suggestion, we set out to identify commonalities between high performing gRNAs using the PAM-flexible NNG SlugCas9 variant described in Figure 4. In this updated version of the manuscript, we are excited to describe a novel finding that (1) the frequency of GFP positive variants positively correlated with the position-weighted free energy change of R-loop formation as estimated by the publicly available CRISPRoff pipeline³ (Figure 4c), and that (2) the frequency of GFP positive variants is a useful proxy for total diversity generated by NNG-SlugCas9-PolI5MΔ (Figure 4d,g). Paired with these findings, we now include in our manuscript the general recommendation that gRNAs include 5' terminal guanines for control of the transcriptional start site and to maximize transcription levels. Additionally, we recommend that gRNAs be designed with 20-21 nt of complementarity to the target site depending on which is predicted to have a more favorable free energy change. We thank Reviewer 1 for inspiring us to seek out and identify this previously obscured design principle.

Minor Comments

-While the PAM-flexible nickases are certainly an engineering improvement, amplicon sequencing of the target region should be performed with the PAM-flexible nickases before any conclusions are drawn about the mutational efficacy of these chimeras compared to enCas9. A direct comparison of the mutation rate and spectrum is highly recommended.

We now include sequencing data for NNG-SlugCas9-PolI5MΔ to validate its capacity to diversify BFP beyond C199 (Figure 4g) and characterize the mutation spectrum (Figure 4f). Of course, per the findings outlined in our manuscript the mutation rates and spectrum will be specific to individual gRNAs and are not meant to represent EvolvR's mutation rates and spectrum in general.

-The introduction should include a discussion of the TRACE targeted mutagenesis system (with a direct reference), especially with respect to the following point.

Though we originally considered discussing and referencing TRACE in our introduction, we opted to keep our introduction focused on diversifiers with the capacity to diversify native genes without the need for introducing an ectopic engineered transgene first. As TRACE requires installing a transgene downstream of a T7 promoter, it was not included. Since our initial submission of the manuscript, the same lab that originally described TRACE reported the use of helicase assisted continuous evolution (HACE), whereby a helicase-deaminase fusion protein is targeted by a nick to generate transition mutations within a 1-2 kb window (Chen et al. 2024⁴). Because HACE is an improvement over TRACE in that it targets native genes without the need for prior genetic engineering with similarly long mutation windows, we believe Reviewer 1's comment is satisfied by our newly introduced reference to HACE in the introduction.

-The limited targeting region of EvolvR limits its applicability for evolving large genes or metabolic pathways – applications suggested in the conclusions that appear unlikely. Practical use of EvolvR necessitates prior knowledge about where to direct the editor, making it less useful for exploratory evolution experiments in native genomic contexts. Most particularly, evolving large genes or gene clusters (e.g., metabolic networks as posed in the conclusion) will be challenging since sgRNA tiling will be required.

We appreciate this nuanced perspective on EvolvR's broad applicability, and now include a discussion of EvolvR's limitations in the manuscript's conclusion. This discussion includes an acknowledgment of EvolvR's limited mutation window and makes a call for identifying or engineering higher processivity error-prone polymerases.

-Lines 92–94 suggest the adenoviral evolution system has the same cheating problem as VEGAS. As far as I know, that is not the case.

Reviewer 1 is correct that the adenoviral evolution system has not been shown to be confounded by cheaters as has been reported for VEGAS. We now explicitly single out VEGAS as having this problem to prevent the reader from inferring that both platforms suffer from it.

-Introducing Map2K1 with 1–2 sentences would help readers understand why it is important etc. We have added 1-2 sentences contextualizing the use of MAP2K1 as a target for directed evolution in mammalian cells.

-A discussion of CRISPR NHEJ methods that have been shown to introduce transversions in vivo in mammalian cells at endogenous loci could help clarify the practical advance over existing technologies.

To our knowledge, NHEJ is a relatively error-free mechanism of blunt-end double stranded DNA repair, which in cooperation with DNA resection and MMEJ can install insertions and deletions at the site of double-stranded breaks. We are not aware of it being leveraged to diversify genomic loci for directed evolution.

-DNA sequences should be included in the Supplemental Information.

We have included annotated DNA sequence tables in the supplemental information.

-Depositing the plasmids in Addgene would significantly amplify the accessibility of this research.

We intend to deposit the plasmids onto Addgene upon publication of the work.

-Lines 357-8 need a reference for PAM flexible Slug-Cas9

We have added the relevant reference.

-Flow cytometry gating examples and raw data should be provided as supplementary information.

We have included flow cytometry gating examples in Supplementary Data Fig. 3. To maximize the likelihood of finding GFP positive cells in conditions where they are present, we have updated our gating strategy to analyze the fluorescence of all cells without excluding doublets. As our raw data files contain many gigabytes of data and are prohibitively large for sharing over the submission portal, we have submitted source data for dot plots including flow cytometry data.

Reviewer #2 (Remarks to the Author):

CRISPR-guided DNA polymerases enable diversification of all nucleotides in a tunable window in *E. coli*, the authors of this paper tested whether EvolvR can diversify targeted genomic loci in human cells. They demonstrated EvolvR's ability to generate transition & transversion mutations across ≥ 50 bp, evolving novel drug-resistant MAP2K1 variants via substitutions unattainable by deaminases. Moreover, they found that EvolvR's mutation window and substitution biases are influenced by the gRNA design and nickase properties, they incorporated a PAM-flexible nickase into EvolvR to make an efficient genetic diversifier. Nonetheless, I expect the authors to conduct a few more additional characterizations to improve the manuscript for eventual publication in Nature Communications.

We thank the reviewer for this summary and for the suggestions that we address below.

Major Suggestions:

1. It has been reported in published literature that EvolvR mainly mutates the A and T bases in the target area in *E. coli*. I think it is necessary to describe the base preference of EvolvR in mammalian cells in detail.

As our manuscript highlights the gRNA-dependent nature of EvolvR's substitution biases, we would expect there to be significant differences in the base preference of substitutions depending on the target sequence, and therefore do not feel comfortable commenting on differences between substitution biases seen in our initial experiments in *E. coli* and those seen in our current work. However, in acknowledgment of the value of showing that EvolvR can generate both transition and transversion mutations, we have now included a heatmap detailing the relative proportion of each substitution type within the set of variants detected by Illumina sequencing for both enCas9-PolI5M Δ and NNG-nSlucCas9-PolI5M Δ (Figure 1g, Figure 4f).

2. In Figure 3, the authors used 18 nt-long gRNAs truncated by 2 nt at their 5' end and assessed enCas9-PolI5M activity according to prior work in *E. coli*, while in mammalian cells, could more sgRNAs with different numbers of truncations not only 2 nt to be tested, to further enhance the mutation efficiency of EvolvR?

We appreciate and share the reviewer's curiosity regarding the extent of truncation that enhances EvolvR's mutagenicity. With respect to Figure 3, we strictly aimed to show that truncated

gRNAs could significantly enhance the diversity of edits with one particular gRNA to demonstrate the role of mismatch tolerance in EvolvR mutagenesis, without asserting that a particular gRNA length is generally optimal (the optimal length likely to vary depending on gRNA sequence and the choice of nuclease considering the thermodynamics underlying R loop formation, as newly outlined in Figure 4 and Supplementary Fig. 10 of this revised manuscript). In general, gRNAs are known to work more or less efficiently at 18 vs 20 bp depending on gRNA sequence (Fu et al. 2014⁵), and therefore we do not recommend gRNA truncation as a means of improving EvolvR's efficiency in general. For the PAM flexible version of EvolvR using NNG-nSlugCas9, we do demonstrate here that gRNAs work less consistently when complementarity with the target site is under 20 bp and rarely generate substitutions consistently with less than 19 bp of complementarity (Supplementary Fig 8).

3. In Figure 3e, the authors found truncated gRNA reduces nCas9-PolI5M-induced indel frequency vs. full-length gRNA, However, only two repeated experiments are not enough to support the conclusion, and the number of repeated experiments needs to be increased.

Given the large effect size of truncating the gRNA, biological duplicates in this case were sufficient to distinguish a statistically significant difference between the truncated and full-length gRNAs using an unpaired student's t test.

4. Whether the off-target effects of the CRISPR component in EvolvR potentially elevate the incidence of false positive results during mutation screening?

For off-target effects stemming from the "CRISPR" (or the nickase) component of EvolvR to result in false positives, there would need to be off-target target sequences in the genome with high similarity to those targeting the gene of interest that by coincidence happen to confer similar functionality as that being selected for in the directed evolution campaign. As this seems highly unlikely to occur, our characterization of EvolvR's off-target mutagenesis focused on quantifying gRNA-independent (or "CRISPR-independent") off-target mutations by coexpressing EvolvR with a gRNA targeting a different locus or one which is inactivated by truncation (see Figures 1d and 3b respectively). Neither of these generated GFP positive cells at levels detectable by our flow cytometry assay. Furthermore, EvolvR's capacity to identify drug-resistant MAP2K1 variants, which were independently validated in a clean genetic background (Figure 2d), demonstrates that our top hits were not the result of off-target mutagenesis.

Minor Suggestions:

5. The author should describe the window size of mutations generated by EvolvR, and what is the maximum range?

We have included an acknowledgment of EvolvR's approximately 40-50 bp-long peak substitution window, though in our sequencing data EvolvR's substitution seems to extend sparsely through the end of the 150 bp-long amplicon sequencing read.

6. The resolution of all figures needs to be improved.

We have increased the resolution of all figures and increased font sizes where possible to improve legibility.

7. In conclusion part, the limitations of EvolvR and possible solutions should be added.

We have included a paragraph in our conclusion outlining EvolvR's biggest limitations and potential solutions.

Thank you for your careful review of our manuscript. We hope you agree that our updates in response to your comments have considerably improved the manuscript.

Reviewer #3 (Remarks to the Author):

Reviewer #4 (Remarks to the Author):

References for citations in responses to reviewers:

1. Ravikumar, A., Arzumanyan, G. A., Obadi, M. K. A., Javanpour, A. A. & Liu, C. C. Scalable, Continuous Evolution of Genes at Mutation Rates above Genomic Error Thresholds. *Cell* **175**, 1946-1957.e13 (2018).
2. Rix, G. *et al.* Scalable continuous evolution for the generation of diverse enzyme variants encompassing promiscuous activities. *Nat. Commun.* **11**, 5644 (2020).
3. Alkan, F., Wenzel, A., Anthon, C., Havgaard, J. H. & Gorodkin, J. CRISPR-Cas9 off-targeting assessment with nucleic acid duplex energy parameters. *Genome Biol.* **19**, 177 (2018).
4. Chen, X. D. *et al.* Helicase-assisted continuous editing for programmable mutagenesis of endogenous genomes. *Science* **386**, eadn5876.
5. Fu, Y., Sander, J. D., Reyon, D., Cascio, V. M. & Joung, J. K. Improving CRISPR-Cas nuclease specificity using truncated guide RNAs. *Nat. Biotechnol.* **32**, 279–284 (2014).